# The Role of Erythropoietin in Metabolic Regulation

**DOI:** 10.3390/cells14040280

**Published:** 2025-02-14

**Authors:** Weiqin Yin, Constance T. Noguchi

**Affiliations:** Molecular Medicine Branch, National Institute of Diabetes and Digestive and Kidney Diseases, NIH, Bethesda, MD 20892, USA; weiqin.yin@nih.gov

**Keywords:** EPO, EPOR, metabolism, red cells

## Abstract

Erythropoietin (EPO) is a key regulator of erythrocyte production, promoting erythroid progenitor cell survival, division, and differentiation in the fetal liver and adult bone marrow. Mice lacking EPO or its receptor (EPOR) die in utero due to severe anemia. Beyond hematopoiesis, EPO influences non-hematopoietic tissues, including glucose and fat metabolism in adipose tissue, skeletal muscle, and the liver. EPO is used to treat anemia associated with chronic kidney disease clinically and plays a role in maintaining metabolic homeostasis and regulating fat mass. EPO enhances lipolysis while inhibiting lipogenic gene expression in white adipose tissue, brown adipose tissue, skeletal muscle, and the liver, acting through the EPO-EPOR-RUNX1 axis. The non-erythroid EPOR agonist ARA290 also improves diet-induced obesity and glucose tolerance providing evidence for EPO regulation of fat metabolism independent of EPO stimulated erythropoiesis. Therefore, in addition to the primary role of EPO to stimulate erythropoiesis, EPO contributes significantly to EPOR-dependent whole-body metabolic response.

## 1. Introduction

Erythropoietin (EPO) is a crucial regulator of erythroid progenitor cells, promoting their growth, survival, and differentiation, and acts by binding to the cell surface erythropoietin receptor (EPOR) to activate downstream signaling pathways. While the primary role of EPO is to regulate red blood cell production, increasing attention has been given to the broader functions of the EPO-EPOR axis, particularly in non-hematopoietic tissues [1,2,3]. Studies have shown that transgenic mice overexpressing human EPO (Tg6) exhibit significantly reduced body weight [4]. In contrast, mice lacking EPOR in adipose tissue display increased body weight and fat mass compared to wild-type controls [5,6]. This suggests that EPO-EPOR signaling influences glucose and fat metabolism. Enhanced EPO signaling in Tg6 mice correlates with improved metabolic regulation, while loss of EPO signaling in non-hematopoietic tissues, as seen in ΔEPORE mice with erythroid restricted EPOR, impairs these processes [6,7].

Recent findings revealed that EPO impacts adipose tissue, skeletal muscle, and the liver by regulating the expression of the transcription factor RUNX1 through EPOR signaling [6]. RUNX1 is a key regulatory protein in vertebrates, known for its role in controlling hematopoiesis and angiogenesis [8]. Beyond hematopoietic functions, RUNX1 participates in regulating the proliferation of mouse epithelial hair follicle stem cells, human epithelial cancer cells, and lipid metabolism. RUNX1 can drive lipid composition changes in hair follicle stem cells by inhibiting lipase activity, thereby reducing the proliferation of mouse epithelial cells and human squamous cell carcinomas [9]. Despite these advances, the precise mechanisms by which EPO-EPOR signaling modulates glucose and lipid metabolism via RUNX1 remain unclear. Understanding this pathway is critical for identifying therapeutic opportunities. This review explores the tissue-protective roles of EPO and its functions in lipid metabolism regulation. We also highlight the emerging EPO-EPOR-RUNX1 axis in metabolic disorders, such as obesity, diabetes, and related conditions. A non-erythropoietic EPOR agonist ARA290 activates EPOR signaling and associated energy-sensing networks providing further evidence of metabolic response to EPO stimulation in non-erythroid tissue. Therefore, in addition to its requisite role in determining erythroid response, endogenous and exogenous EPO-EPOR signaling contributes importantly to energy homeostasis.

## 2. EPO Treatment in Chronic Kidney Disease

Erythropoietin (EPO) is the primary hormone regulating erythrocyte production and is produced in the adult kidney mainly by renal perivascular cells, predominately interstitial fibroblasts. However, in the injured kidney or chronic kidney disease, these cells can differentiate into myofibroblasts resulting in fibrosis and a decrease in endogenous EPO production leading to anemia [10]. Anemia in chronic kidney disease (CKD) due to decreased endogenous kidney EPO production is also related to dysregulation of iron metabolism, blood loss, inflammation, and other processes that are associated with increased morbidity and adverse clinical outcomes [11]. The current standard treatment for renal anemia addresses both EPO deficiency and iron imbalance. In addition to administration of recombinant human EPO, orally active hypoxia-inducible factor prolyl-hydroxylase domain inhibitors that increase endogenous EPO production can be used to increase serum EPO concentrations to improve anemia of patients with kidney disease not on dialysis and patients with chronic kidney disease [12,13]. In addition to managing renal anemia, increasing EPO levels may improve blood glucose levels and glucose tolerance [14,15] and potentially affect metabolism in CKD patients.

Since receiving approval from the U.S. Food and Drug Administration in 1989 for the treatment of anemia in chronic kidney disease, recombinant human EPO has been used for more than three decades for the treatment of renal anemia due to insufficient endogenous EPO production in patients with kidney disease and dialysis patients with chronic kidney disease [3,16,17,18,19]. EPO use in kidney disease relieves anemia symptoms, avoids blood transfusions, and was reported to improve the patient’s quality of life and exercise ability, but in some patients was associated with hypertension and elevated blood pressure [16]. There is no clear indication of the impact of EPO treatment on human fat metabolism. 

## 3. EPO Activity Beyond Erythropoiesis

Animal models have been instrumental in uncovering the roles of EPO-EPOR signaling in both erythroid and non-erythroid tissues. Studies using genetically modified mice with targeted deletions of EPO or EPOR demonstrated that while these factors are not essential for primitive erythropoiesis, they are critical for definitive fetal liver and adult erythropoiesis. Mature red blood cell formation requires EPO and EPOR for the survival, proliferation, and differentiation of maturing erythroid progenitor cells [20,21]. In adults, EPO, produced by interstitial cells in the kidney, is inducible under hypoxic conditions and increases in response to anemia, high altitude, or ischemic stress [22,23,24]. Additionally, animal models have revealed that EPOR expression extends beyond hematopoietic and erythroid cells to select non-hematopoietic tissues. This broader distribution of EPOR expression enables EPO to exert diverse physiological effects, including responses to ischemic and traumatic brain injury, neurocognitive recovery, wound healing, bone remodeling, and the regulation of metabolic homeostasis (Figure 1) [1,3,25]. Expression of EPOR and EPO production in mouse brain indicates EPO activity on the other side of the blood–brain barrier that contributes to neural cell proliferation, viability, and protection against ischemia and glutamate damage [26,27,28,29,30]. In rodent models of diabetes, EPO treatment contributed to improved cognitive dysfunction associated with hippocampal neurodegeneration [31,32]. Potential benefit in the nervous system of EPO treatment is suggested by clinical trials in patients with ischemic stroke and other neurologic disorders [33,34,35]. Long-term darbepoetin-α treatment in the ApoE-/- mouse model for atherosclerotic disease reduced oxidative stress that is characterized by lipid profile, inflammation, endothelial injury, lipid peroxidation, and protein oxidation [36].

During development, EPOR expression in non-hematopoietic tissue includes heart and brain and EPOR knockout mice exhibit brain and heart hypoplasia prior to death at embryonic day 13.5, adding further support for direct non-hematopoietic tissue response to EPO [1,37,38,39]. Fetal liver erythropoiesis is supported directly by EPO production in the fetal liver. At birth, EPO production shifts from the fetal liver to the adult kidney while hepatocytes retain the ability to produce EPO in a hypoxia-dependent manner, albeit at a much lower level than the kidney. Hepatocyte EPO production cannot compensate for the reduction in EPO production by the kidney in chronic kidney disease [40]. EPO directly impacts liver health in animal models. EPO treatment or increasing EPO signaling exhibit protective activity in animal models of liver ischemia-reperfusion injury associated with transplantation including bone marrow-derived mesenchymal stem cell transplantation in a mouse liver fibrosis model [41,42,43,44]. 

## 4. Regulation of EPOR Expression

Erythroid cells express the highest levels of EPOR during the differentiation of erythroid progenitor cells, where EPOR is essential for their survival, proliferation, and differentiation. Mice lacking EPO or EPOR die mid-gestation due to severe anemia [20,21,45]. EPOR expression is regulated at the transcriptional level by erythroid-specific transcription factors GATA1 and TAL1, which bind to the EPOR 5′ promoter and 5′ untranslated transcribed region [46,47,48]. EPO itself regulates EPOR expression, as EPO stimulation in erythroid progenitor cells increases the expression of GATA1 and TAL1, which in turn transactivate EPOR expression. However, EPOR is downregulated and undetectable in terminally differentiated reticulocytes or mature erythroid cells [45]. EPO binds to the EPOR homodimer complex on the surface of erythroid cells, activating JAK2 associated with the EPOR cytoplasmic domain. This interaction triggers downstream signaling pathways, including STAT5 activation [49,50,51]. EPOR expression is not limited to erythroid tissues, as GATA- and TAL1-binding motifs in the 5′ region of the EPOR gene also contribute to its expression in non-erythroid cells [25]. For example, EPOR expression in skeletal muscle satellite cells and myoblasts is regulated in part by transcription factors GATA3 and GATA4 that bind to the GATA motif, and by TAL1, MyoD, and Myf-5 that bind to the E-BOX TAL1-binding motifs and mediates EPO stimulated proliferation and survival [52,53,54]. In mice, increased EPO signaling enhanced skeletal muscle repair and recovery, increased maximum load tolerated by skeletal muscle, and improved survival of transplanted skeletal muscle myoblasts [55,56].

The availability of recombinant human EPO and its use in animal models has facilitated the identification of EPOR-mediated EPO responses in non-erythroid tissues. These include vascular endothelium and the cardiovascular system, neural cells and the brain, myoblasts and skeletal muscle, adipose tissue, and bone marrow stromal cells [1,3]. 

## 5. Effects of EPO on Glucose Metabolism

EPO treatment in mice increases hematocrit and improves glucose tolerance on both normal chow and high-fat diets, and in both male and female mice [3]. In contrast, EPO regulation of body weight and fat mass readily observed in male mice exhibits a sex-differential response related to the anti-obesity activity of estrogen in female mice [57]. The glucose intolerance observed in male and female mice lacking estrogen receptor α was improved by EPO treatment, independent of changes in body weight and fat mass [58]. In male mice, particularly those on a high-fat diet, EPO treatment increased WAT oxidative metabolism, fatty acid oxidation, brown fat-associated gene expression, mitochondrial content, and uncoupled respiration, mediated by peroxisome proliferator-activated receptor (PPAR) α and SIRT1 activation [5]. In contrast, mice with targeted deletion of EPOR in adipose tissue on a high-fat diet showed decreased glucose tolerance, increased body and fat mass, elevated serum leptin and insulin levels, and reduced expression of brown fat-associated genes, mitochondrial activity, and AKT activity that did not respond to EPO stimulation [5]. EPO administration in high-fat diet-fed mice improved glucose metabolism and insulin resistance, mediated in part by PPARγ and SIRT1 activity and the PI3K/AKT pathway in the liver [59]. Other tissues that contribute to EPO activity to improve obesity and glucose homeostasis in obese mice include brown adipose tissue (BAT), in which EPO increased interscapular fat mass, temperature, and secretion of FGF21, a metabolic regulator of glucose homeostasis and insulin sensitivity [60].

In young male mice, elevated EPO via transgenic expression or EPO treatment improved glucose tolerance and insulin sensitivity, along with changes in lipid metabolism-associated gene expression [6]. In a type 1-like diabetic mouse model, elevated EPO treatment improved blood glucose in a dose-dependent manner, increased insulin sensitivity, and reversed the decreased GLUT4 levels in skeletal muscle [61]. EPO treatment in male rats on a high-fat diet improved glucose tolerance without changing body weight or fat mass. Furthermore, EPO administration in a streptozotocin diabetic rat model improved glucose tolerance, insulin tolerance, pancreatic β-cell dysfunction, and increased antioxidant activity [62,63]. In a mouse model of non-alcoholic fatty liver disease, EPO administration reduced body weight, reversed glucose intolerance and insulin resistance, reduced lipid accumulation in the liver and WAT, suppressed lipid synthesis genes in the liver, and increased lipolysis proteins in adipose tissue [64]. In a female rat model of polycystic ovary syndrome with impaired glucose tolerance and elevated plasma insulin, EPO treatment improved glucose tolerance, decreased serum insulin levels, and reduced disease-associated reproductive manifestations, although to a lesser extent than metformin [65]. In domestic pigs, transgenic EPO expression lowered fasting glucose levels and improved glucose tolerance on both normal chow and high-fat diets [66]. 

While elevated brain EPO did not increase hematocrit due to protection by the blood–brain barrier, improvement in glucose metabolism was observed. Increased brain EPO in mice, via transgenic expression or intracerebral ventricular EPO pump, improved glucose tolerance in both male and female mice on normal chow and high-fat diets, independent of body weight or fat mass [3]. Hypothalamic ventricular administration of EPO in aged obese mice decreased food intake and obesity, improved metabolic function, and reversed impairments in glucose tolerance and insulin sensitivity without increasing hematocrit [67]. These animal models demonstrate that EPO treatment improves glucose metabolism independent of gender, changes in body weight or fat mass, and increased hematocrit from EPO-stimulated erythropoiesis. 

## 6. EPO Regulation of Fat Mass and Metabolic Homeostasis

While EPO functions primarily to regulate erythropoiesis by binding to erythroid cell surface EPOR, EPOR expression extends beyond erythroid progenitor cells, enabling non-erythroid EPO activity [2,3]. ΔEPORE mice that express EPOR only in erythroid tissue become glucose intolerant, insulin resistant, and obese with age, providing evidence that endogenous EPO activity plays a role in regulating metabolic homeostasis [7]. In a rat model of chronic renal failure, increased plasma cholesterol concentration was lowered by EPO treatment while LDL-R expression was unchanged [68]. High EPO expression in skeletal muscle, achieved via gene transfer in mice, protected against obesity, reduced fat mass and body weight, and normalized glucose tolerance and fasting insulin levels in high-fat diet-fed mice [69]. Increased EPO levels by transgenic expression or EPO administration reduced blood glucose in both obese and non-obese mice, and lowered body weight in male obese and diabetic mice [4,70]. EPO treatment in male obese mice improved glucose tolerance and insulin sensitivity, reduced body weight, and decreased lipid storage in WAT and the liver. In a mouse model of non-alcoholic fatty liver disease, EPO administration in high-fat diet-induced obese mice reduced body weight, improved glucose tolerance and insulin sensitivity, and reduced lipid accumulation in liver and white adipose tissue (WAT) [64]. EPO suppressed proteins associated with lipid synthesis in the liver, and increased lipolysis proteins in visceral WAT, mediated via STAT3 and STAT5 activation.

EPO treatment modulated lipid metabolism gene expression by decreasing lipogenic genes and increasing lipolytic genes [6,64,69]. Increased EPO through intermittent hypobaric hypoxia improved glucose metabolism, as well as iron and lipid metabolism in rodents [71,72]. The obese phenotype observed in mice with EPOR deletion in non-erythroid tissues, particularly in adipose tissue, highlights the role of endogenous EPO in metabolic regulation [5,7]. This contrasts the lean phenotype and improved glucose tolerance observed with increased EPO by transgene expression, EPO administration, or hypoxia [4,69,73].

Beyond EPOR expression in erythroid progenitor cells, EPOR is expressed at relatively high levels in WAT on adipocytes and stromal vascular cells [74]. ΔEPORE mice with EPOR restricted to erythroid tissue exhibit decreased energy expenditure and total activity, an age-dependent increase in obesity, and a decrease in glucose tolerance and insulin sensitivity, suggesting that EPO signaling in non-erythroid tissue promotes a lean phenotype [7,75]. In wild-type mice, elevated EPO decreased fat mass accumulation with an increase in mitochondrial oxidative gene expression, an increase in brown fat-associated gene expression, and a decrease in expression of white fat-associated genes in WAT in male mice and ovariectomized female mice [4,5,57]. EPO treatment in male mice increases AKT activation in WAT which is also observed with insulin stimulation, and increases mitochondrial biogenesis and cellular respiration rate mediated via EPOR expression [5]. Targeted deletion of EPOR in adipose tissue in mice increases susceptibility to obesity, decreases glucose tolerance, and increases insulin resistance [5]. PPARα cooperates with Sirt1 to mediate WAT EPO activity to induce brown fat-like gene expression including PGC-1α and uncoupled respiration [5]. In addition to the modification of adipocyte response to EPO by genetic manipulation of EPOR expression, mouse background strain may also affect EPO regulation of glucose tolerance, insulin sensitivity, and fat mass accumulation [5,76]. EPO treatment in female mice demonstrated the expected increase in hematocrit without changes in fat mass [57]. This sex-specific EPO regulation of fat mass is due to the protective effect of estrogen on obesity in female mice that contributes to energy metabolism, insulin sensitivity, and lipid metabolism [57,77]. Estrogen supplementation in ovariectomized female mice abrogates the decrease in fat mass with EPO administration [57]. Targeted deletion of estrogen receptor α by whole-body knockout or by an adipose tissue-specific knockout reduces the protective effect of estrogen on obesity. During high-fat diet feeding, EPO treatment increases hematocrit and also exhibits an anti-obesity effect in these female mice, decreasing fat mass accumulation and increasing brown fat-associated gene expression in WAT [58]. While EPO stimulation of erythropoiesis and increased hematocrit is gender neutral, the sex-dimorphic EPO regulation of fat mass is dependent on estrogen and estrogen receptor α response in adipose tissue. EPO activity in BAT of young male mice on the high-fat diet was also suggested to stimulate STAT3 signaling, upregulate brown fat-associated transcription factor PRDM16 and mitochondria-associated total uncoupling protein 1 (UCP1), increase oxygen consumption, improve glucose tolerance, and reduce body weight and fat mass [60].

In humans, full-heritage PIMA Indians are associated with a high incidence of obesity and type 2 diabetes [78,79], and a subset analysis showed that endogenous plasma EPO level exhibited an inverse association with percent weight change per year among males that was not evident in females and was not dependent on EPO regulation of erythropoiesis [80]. EPO regulation of metabolism combined with EPO induction by hypoxia-inducible factor (HIF) that gives rise to increased erythropoiesis and hemoglobin at high altitude may account for the reduced incidence of obesity associated with military recruits (about 94% male) in the United States stationed at high altitude [81,82,83,84].

## 7. EPO Regulation of Lipolytic and Lipogenic Gene Expression 

Increased EPO signaling in WAT and adipocytes, mediated by elevated EPO binding to its receptor, promotes the expression of mitochondrial biogenesis genes, the key metabolic regulator PGC-1α, and brown fat-associated genes such as UCP1, and enhances oxygen consumption [5,6]. EPO stimulation also modulates lipid metabolism gene expression, promoting a lean phenotype by increasing the expression of lipolytic genes and suppressing the expression of lipogenic genes [6]. In the subcutaneous WAT (scWAT) of transgenic mice with high EPO expression and EPO-treated wild-type mice, increased EPO activity resulted in the upregulation of lipolytic genes, PPARγ and lipoprotein lipase (Lpl), along with downregulation of lipogenic genes such as acetyl-CoA carboxylase 1 (Acc1), acetyl-CoA carboxylase 2 (Acc2), Fas receptor (Fas), Lipin1, sterol regulatory element binding transcription factor 1 (Srebf1), and stearoyl-CoA desaturase 1 (Scd1). In epididymal WAT, transgenic mice with high EPO did not show increased expression of brown fat-associated genes; however, both transgenic mice with high EPO and EPO-treated mice showed increased expression of lipolytic genes and decreased expression of lipogenic genes. Similar changes in lipid metabolism gene expression were observed in corresponding BAT, skeletal muscle, and liver in mice with elevated EPO [6]. In contrast, in mice with EPOR restricted to erythroid tissue and in mice with targeted deletion of EPOR in adipose tissue, EPO treatment resulted in the expected increase in hematocrit but no change in lipid metabolism gene expression in subcutaneous or epididymal WAT, BAT, skeletal muscle, or liver. The lack of EPO regulation of lipid metabolism gene expression in these tissues suggests that the ability of EPO to regulate lipid metabolism and promote a lean phenotype is largely mediated by non-erythroid EPO activity, particularly by EPOR expression in WAT, and is independent of EPO-stimulated erythropoiesis [6].

## 8. Activation of Non-Erythroid EPOR 

ARA290, or Cibinetide, a peptide derived from the structure of erythropoietin but without erythropoietic activity, is proposed to bind to an EPO receptor heterodimer composed of EPOR and the β common receptor (CD131) in non-hematopoietic tissue, promoting tissue-protective activity associated with EPO [85,86]. Like EPO, ARA290 improved glucose tolerance and insulin release in diabetic rats and enhanced immune function while reducing disease activity in a mouse model of dextran sulfate sodium-induced colitis [87,88]. The anti-inflammatory response to ARA290 improved mouse models of systemic lupus erythematosus and ischemic retina. In these models, ARA290, but not EPO, reduced the pro-inflammatory response and decreased avascular areas [89,90]. Unlike high-dose EPO treatment, which increases hematocrit and promotes bone loss, ARA290 increases bone mineral density and reduces osteoclast generation [91]. In patients with sarcoidosis, ARA290 improved neuropathy and increased corneal nerve fibers [92,93]. Additionally, ARA290 improved neuropathy and metabolic profiles in patients with type 2 diabetes and enhanced glucose tolerance and insulin release in diabetic rats [87,93,94]. Administration of ARA290 in male mice decreased body weight and fat mass, improved glucose tolerance, and, unlike EPO treatment, did so without increasing erythropoiesis or hematocrit levels [6]. ARA290 mimicked EPO regulation of lipid metabolism gene expression by increasing lipolysis genes and decreasing lipogenic genes in subcutaneous and epididymal WAT, BAT, skeletal muscle, and liver. In male mice on a high-fat diet, ARA290 treatment promoted an even greater metabolic response, with an increase in lipolytic gene expression, a decrease in lipogenic gene expression, and improved body weight, fat mass, and glucose tolerance [6]. EMP9 is an EPOR antagonist that lacks proliferative activity and is derived from EMP1, an EPO mimetic peptide that promotes dimerization of EPOR and proliferation of EPO-responsive cells [95]. Unlike ARA290 administration, treatment with EMP9 in male mice increased lipogenic gene expression and decreased lipolytic gene expression in WAT, BAT, skeletal muscle, and liver, with no change in glucose tolerance. The improvement in body weight and fat mass, and the changes in lipid metabolism gene expression with ARA290 treatment in mice on normal chow or high-fat diet were abrogated by combined treatment with ARA290 and EPOR antagonist EMP9 [6].

## 9. Regulation of Energy Metabolism Through the EPO-EPOR-RUNX1 Axis

As a transcription factor, RUNX1 is a key regulatory protein in vertebrates. It plays a crucial role in blood formation and vascular development. During embryogenesis, RUNX1 is involved in the emergence of hematopoietic sites, and RUNX1 is essential for the differentiation of bone marrow hematopoietic stem cells in adults [96]. Beyond its role in the blood system, RUNX1 also influences lipid metabolism. It can affect the proliferation of mouse skin epithelial cells and human skin and oral squamous cell carcinoma by modifying the activity of lipid enzymes such as Scd1 and Soat1 [9]. In lipid metabolism, RUNX1 regulates gene expression by binding to promoters of lipogenic and lipolytic genes that contain a RUNX1 binding motif (Figure 2) [6]. In the liver, CBFA2/RUNX1 partner transcriptional co-repressor 3 (CBFA2T3) can be activated by PPARα, influencing glucose tolerance and insulin resistance in mice [97].

In mice, elevated EPO increases RUNX1 protein stability in scWAT through activation of EPO-EPOR signaling [6]. The 5’ flanking regions of lipolytic and lipogenic genes in scWAT that respond to EPO stimulation to promote a lean phenotype contain RUNX1 binding motifs. In reporter gene assays, these RUNX1 binding motifs from lipolytic genes exhibit RUNX1-dependent enhancer activity, while those from lipogenic genes exhibit RUNX1-dependent silencer activity [6]. Mice expressing high transgenic EPO treated with the RUNX1 inhibitor Ro5-3335 show a decrease in glucose tolerance, and the improvement in lipid metabolism gene expression associated with elevated EPO is no longer evident in subcutaneous or epididymal WAT, BAT, skeletal muscle, or liver [6].

Core binding factor β (CBFβ) partners with RUNX1 to enhance RUNX1 binding to DNA and transcriptional regulation [98,99]. RUNX1 stability is regulated by post-translational ubiquitylation, leading to rapid degradation through the ubiquitin-proteasome pathway [100]. E3 ubiquitin ligases are required for ubiquitination, and in scWAT, elevated EPO-EPOR signaling specifically decreases FBXW7 E3 ubiquitin ligase mRNA and protein, coincident with increased RUNX1 protein stability (Figure 2) [6]. EPO-EPOR signaling affects both RUNX1 and the expression of CBFβ, although increasing CBFβ did not appear to protect RUNX1 from degradation by increased FBW7 [6]. FBW7 protein may regulate adipocyte metabolism via RUNX1 ubiquitylation to increase RUNX1 proteasome degradation, and EPO-EPOR signaling decreases FBW7 expression, resulting in increased RUNX1 stability and RUNX1 transcription activity. This ultimately regulates lipid metabolism by increasing lipolytic gene expression and decreasing lipogenic gene expression to promote a lean phenotype.

EPO administration in mice increased oxidative metabolism, fatty acid oxidation, and lipolytic gene expression, and promoted mitochondrial content and uncoupled respiration in adipose tissue [5,6]. Increased expression of lipolytic genes increases triglyceride catabolism, breaking down stored fat into free fatty acids in adipose tissue and numerous nonadipose tissues, and promoting tissue lipid mobilization and utilization such as energy production for skeletal muscle, and affecting gene transcription, the cell cycle, and cell growth [101]. These activities associated with increased lipid metabolism likely contribute to EPO activity in non-erythroid tissue including enhanced wound healing in skeletal muscle [55,56].

Metabolism coregulators, peroxisome proliferator-activated receptor (PPAR) α and Sirt1, along with RUNX1 combine to mediate, in part, EPO regulation of metabolism, including gene expression associated with lipid metabolism [5,6]. Decreases in fat mass in mice treated with EPO are consistent with EPO-stimulated down-regulation of lipogenic genes that decrease expression of enzymes crucial for fatty acid production, reduce fatty acid synthesis, lipid production, and fat storage [6,7]. EPO-regulated reduction of lipogenic genes could potentially help reduce fat synthesis that is associated with high triglyceride levels in liver and adipose tissues in metabolic disease or obesity [102]. Together, EPO as a novel regulator of energy homeostasis including lipolytic and lipogenic gene expression provides a potential strategy to enhance metabolic health and protect against obesity. 

## 10. Conclusions

In addition to stimulating the production of erythroid progenitor cells, EPO/EPOR also influences fat metabolism and glucose tolerance. Recent studies confirm that the EPO-EPOR-RUNX1 axis plays a critical role in maintaining energy metabolism and homeostasis in mice [6]. EPO administration in mice protected against diet-induced obesity, enhanced energy expenditure, and reduced fat accumulation. This treatment also improved glucose tolerance and alleviated insulin resistance. Conversely, when mice were injected with EPO and RUNX1 inhibitors, they exhibited glucose intolerance and insulin resistance, and developed obesity. Furthermore, the EPO analog ARA290 that activates EPOR in non-erythroid tissue without stimulating erythropoiesis has been shown to positively impact diet-induced obesity and glucose tolerance in mice. While these findings underscore the importance of EPO-EPOR-RUNX1 signaling in fat metabolism, its application to the study of fat metabolism in other organs remains in its early stages. Further research is essential to elucidate the central mechanisms by which EPO-EPOR-RUNX1 regulates fat metabolism across various tissues.

## Figures and Tables

**Figure 1 cells-14-00280-f001:**
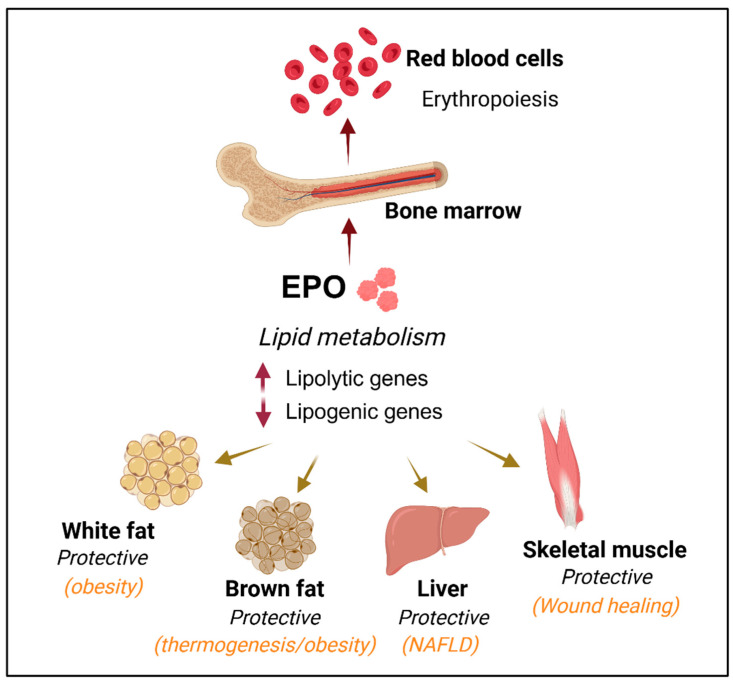
EPO affects fat metabolism in multiple tissues other than hematopoietic tissues. The primary function of EPO is to stimulate the proliferation, survival, and differentiation of bone marrow erythroid progenitor cells and regulate the production of erythrocytes. Animal models provide evidence that non-erythroid EPO activity contributes to metabolism and energy homeostasis. EPO regulates lipid metabolism to increase lipolytic gene expression and suppress lipogenic gene expression to promote a lean phenotype in non-erythroid tissues. These non-erythroid tissues include white adipose tissue (WAT), where EPO decreases fat mass accumulation and susceptibility to diet-induced obesity, brown adipose tissue (BAT), where EPO increases oxygen consumption and contributes to the improvement of glucose tolerance and fat mass, skeletal muscle, where EPO improves muscle maintenance and wound healing, and liver, where EPO decreases fat accumulation associated with non-alcoholic fatty liver disease. Created in BioRender. Yin, W. (2025) https://BioRender.com/r49w251.

**Figure 2 cells-14-00280-f002:**
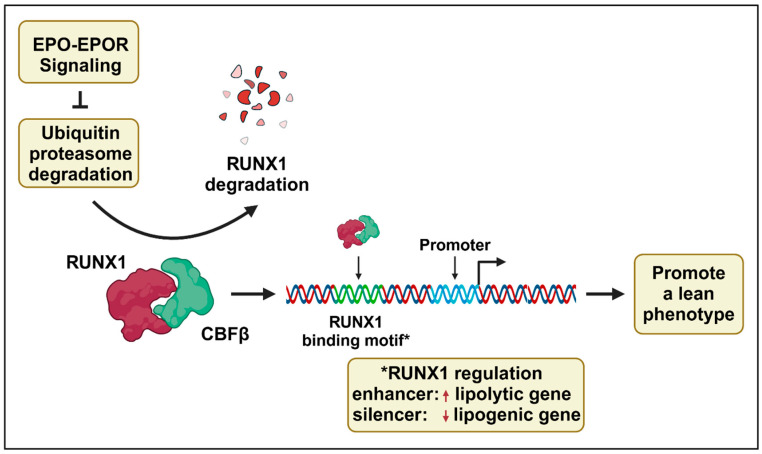
EPO-EPOR-RUNX1 axis affects fat accumulation. The EPO-EPOR signaling pathway stabilizes RUNX1 in subcutaneous white adipose tissue (scWAT) by inhibiting the ubiquitination of RUNX1 which is required for proteosome degradation. RUNX1 partners with CBFβ to form a heterodimer that regulates transcription via RUNX1 binding to DNA. RUNX1 binding motifs in promoters of lipid metabolism genes contribute to a lean phenotype. The RUNX1 binding motifs act as enhancers in lipolytic genes to increase expression and as suppressers in lipogenic genes to decrease expression. Created in BioRender. Yin, W. (2025) https://BioRender.com/h58j247.

## Data Availability

Not applicable.

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
