# Peer review of "The Role of Erythropoietin in Metabolic Regulation"

_cells, 2025, doi:10.3390/cells14040280_

Round 1

Reviewer 1 Report

Comments and Suggestions for Authors

This is a very interesting and well written review on the role of EPO and the EPO R axis for the regulation of fat and glucose metabolism.

 The positive effect of EPO in the streptozotocin diabetic rat model and its anti-oxidant activity may be partly attributed to Epo mediated inhibition of NFkappa B mediated activation of inflammatory effector functions (Nairz M et al. Immunity 2011). This may also contribute to lowering of leptin in adipose tissue which remains to be shown.

Does EPO affect the expression of LDL-R or reverse cholesterol transport? This should be mentioned.

Does treatment with Epo or EPO mimietics alter the lipid profile in the circulation and the risk for atherosclerosis in mouse disease models (f.e. ApoE -/- with high fat diet)?

Under inflammatory conditions HIF-1 is thought to result in re-programming of glucose consumption in favor of aerobic glycolysis via the Warburg pathways (Cheung et al. Science 2014). Is this pathway affected by Epo-mimetics as they may modify HIF stability?

Author Response

Comments 1:

This is a very interesting and well written review on the role of EPO and the EPO R axis for the regulation of fat and glucose metabolism.

 The positive effect of EPO in the streptozotocin diabetic rat model and its anti-oxidant activity may be partly attributed to Epo mediated inhibition of NFkappa B mediated activation of inflammatory effector functions (Nairz M et al. Immunity 2011). This may also contribute to lowering of leptin in adipose tissue which remains to be shown. 

Does EPO affect the expression of LDL-R or reverse cholesterol transport? This should be mentioned.

Does treatment with Epo or EPO mimietics alter the lipid profile in the circulation and the risk for atherosclerosis in mouse disease models (f.e. ApoE -/- with high fat diet)?

Under inflammatory conditions HIF-1 is thought to result in re-programming of glucose consumption in favor of aerobic glycolysis via the Warburg pathways (Cheung et al. Science 2014). Is this pathway affected by Epo-mimetics as they may modify HIF stability?

This is a very interesting and well written review on the role of EPO and the EPO R axis for the regulation of fat and glucose metabolism.

 The positive effect of EPO in the streptozotocin diabetic rat model and its anti-oxidant activity may be partly attributed to Epo mediated inhibition of NFkappa B mediated activation of inflammatory effector functions (Nairz M et al. Immunity 2011). This may also contribute to lowering of leptin in adipose tissue which remains to be shown. 

Does EPO affect the expression of LDL-R or reverse cholesterol transport? This should be mentioned.

Response 1: In a rat model of chronic renal failure, increased plasma cholesterol concentration was lowered by EPO treatment while LDL-R expression was unchanged (Alsaran, Sabry et al. 2009).

Alsaran, K. A., A. A. Sabry, A. H. Alghareeb and G. Al Sadoon (2009). "Effect of hepatitis C virus on hemoglobin and hematocrit levels in saudi hemodialysis patients." Ren Fail 31(5): 349-354.

This is now included in the text in section 6 (EPO regulation of fat mass and metabolic homeostasis).

Comments 2: Does treatment with Epo or EPO mimietics alter the lipid profile in the circulation and the risk for atherosclerosis in mouse disease models (f.e. ApoE -/- with high fat diet)?

Response 2: Long term darbepoetin-a treatment in the ApoE-/- mouse model for atherosclerotic disease reduced oxidative stress that is characterized by lipid profile, inflammation, endothelial injury, lipid peroxidation and protein oxidation (Ozdemir, Hanikoglu et al. 2017).

Ozdemir, E. D., A. Hanikoglu, A. Cort, B. Ozben, G. Suleymanlar and T. Ozben (2017). "Effects of long- and short-term darbepoetin-alpha treatment on oxidative stress, inflammation and endothelial injury in ApoE knockout mice." Adv Clin Exp Med 26(4): 635-643.

This is now included in the text in section 3 (EPO activity beyond erythropoiesis).

Comments 3: Under inflammatory conditions HIF-1 is thought to result in re-programming of glucose consumption in favor of aerobic glycolysis via the Warburg pathways (Cheung et al. Science 2014). Is this pathway affected by Epo-mimetics as they may modify HIF stability?

Response 3: As alluded to by the reviewer, HIF can increase erythropoietin expression while the inflammation associated transcription factor NFkappa B can inhibit erythropoietin expression (Nairz, Sonnweber et al. 2012). Epo and its derivatives can contribute to select tissue protection and immune regulation including suppression of NFkappa B (Peng, Kong et al. 2020) while HIF stability is not directly modified by Epo, However, increased Epo stimulated erythropoiesis increases oxygen delivery to tissues which reduces hypoxic conditions and can indirectly modify HIF stability via oxygen-dependent hydroxylation by HIF prolyl hydroxylases and proteasomal degradation (Semenza 2022).

Nairz, M., T. Sonnweber, A. Schroll, I. Theurl and G. Weiss (2012). "The pleiotropic effects of erythropoietin in infection and inflammation." Microbes Infect 14(3): 238-246.

Peng, B., G. Kong, C. Yang and Y. Ming (2020). "Erythropoietin and its derivatives: from tissue protection to immune regulation." Cell Death Dis 11(2): 79.

Semenza, G. L. (2022). "Breakthrough science: hypoxia-inducible factors, oxygen sensing, and disorders of hematopoiesis." Blood 139(16): 2441-2449.

Reviewer 2 Report

Comments and Suggestions for Authors

The well-written and timely overview of an important subject, i.e. effect of erythropoietin and erythropoietin receptor signaling on non-erythroid tissues written by major contributor to this field, Connie Noguchi.

My only critique is that in many paragraphs similar descriptions and references are made on erythropoietin signaling in adipose tissue. These seem to be quite repetitious; some condensation would be desirable.

Author Response

Comments 1: The well-written and timely overview of an important subject, i.e. effect of erythropoietin and erythropoietin receptor signaling on non-erythroid tissues written by major contributor to this field, Connie Noguchi.

My only critique is that in many paragraphs similar descriptions and references are made on erythropoietin signaling in adipose tissue. These seem to be quite repetitious; some condensation would be desirable.

Response 1: We thank the reviewer for pointing out the repetitions. The text has been edited as suggested.

Reviewer 3 Report

Comments and Suggestions for Authors

Manuscript ID: cells-3406032

Type of manuscript: Review

Title: Erythropoietin (EPO) metabolic regulation and the role of RUNX1-EPO-EPO receptor axis in fat metabolism

Authors: Weiqin Yin, Constance Tom Noguchi *

The review by Yin and Noguchi describes the extra hematopoietic role of erythropoietin (Epo) in regulating fat metabolism. The manuscript overall is very successful. It provides an overview of the role of Epo/EpoR signaling in adipose tissue and its role in glucose and fat metabolism. The review is well balanced and appropriately referenced. It should be a valuable addition to the literature.

I have a few minor comments:

Line 55 should say “is used clinically” or “is used in the clinic”.

Increased production of hepatic epo is seen in CKD patients, is this compensatory response impaired in obese mice/patients? 

Line 240 – what is the ?

The use of ARA290 and EMP9 are interesting reagents in the discussion. Could you provide more information of how each works. In addition, what is know about the differences in signaling when ARA290 v. Epo is used?

Author Response

Comments 1: 

Manuscript ID: cells-3406032

Type of manuscript: Review

Title: Erythropoietin (EPO) metabolic regulation and the role of RUNX1-EPO-EPO receptor axis in fat metabolism

Authors: Weiqin Yin, Constance Tom Noguchi *

The review by Yin and Noguchi describes the extra hematopoietic role of erythropoietin (Epo) in regulating fat metabolism. The manuscript overall is very successful. It provides an overview of the role of Epo/EpoR signaling in adipose tissue and its role in glucose and fat metabolism. The review is well balanced and appropriately referenced. It should be a valuable addition to the literature.

I have a few minor comments:

Line 55 should say “is used clinically” or “is used in the clinic”.

Response 1: 

We thank the reviewer for the suggestion. This phrase is now deleted and EPO use is indicated in the beginning of the next paragraph. “

Comments 2: 

Increased production of hepatic epo is seen in CKD patients, is this compensatory response impaired in obese mice/patients? 

response 2:

Sodium-glucose cotransporter 2 (SGLT2) inhibitors used for weight loss is reported to enhance erythropoiesis by increased erythropoietin production that is of kidney and possibly predominately of liver origin (Packer 2023). SGLT2 effectiveness in patients with type 2 diabetes (Thiele, Rau et al. 2021), suggests that increased production of hepatic Epo seen in chronic kidney disease patients is not impaired in obese patients.

Packer, M. (2023). "Mechanisms of enhanced renal and hepatic erythropoietin synthesis by sodium-glucose cotransporter 2 inhibitors." Eur Heart J 44(48): 5027-5035.

Thiele, K., M. Rau, N. K. Hartmann, J. Mollmann, J. Jankowski, M. Bohm, A. P. Keszei, N. Marx and M. Lehrke (2021). "Effects of empagliflozin on erythropoiesis in patients with type 2 diabetes: Data from a randomized, placebo-controlled study." Diabetes Obes Metab 23(12): 2814-2818.

Comments 3:

Line 240 – what is the a?

Response 3:

In our error, words were deleted from the sentence. The sentence has been corrected and now reads: Targeted deletion of estrogen receptor a by whole body knockout or by adipose tissue specific knockout reduces the protective effect of estrogen to obesity.

Comments 4:

The use of ARA290 and EMP9 are interesting reagents in the discussion. Could you provide more information of how each works. In addition, what is know about the differences in signaling when ARA290 v. Epo is used?

Response 4:

The two sections that deal with ARA290 have been combined for clarity. As indicated in the text, “ARA290, or Cibinetide, a peptide derived from the structure of erythropoietin but without erythropoietic activity, is proposed to bind to an EPO receptor heterodimer composed of EPOR and the β common receptor (CD131) in non-hematopoietic tissue, promoting tissue-protective activity associated with EPO.”

As suggested by the reviewer, we also now include in the text that EMP9 is an EPOR antagonist that lacks proliferative activity and is derived from EMP1, an Epo mimetic peptide that promotes dimerization of EpoR and supports proliferation of Epo-responsive cells (Johnson, Farrell et al. 1998). EMP9 contains a substitution of alanine for glycine in the center of EMP1.

The activities of Epo and ARA290 share some similarities and differences and appear to depend on the experimental system studied.  For example, in an in vitro model for oxygen induced retinopathy of the ischemic retina, Epo and ARA290 activated PI3K signaling and transient STAT3 phosphorylation of endothelial colony forming cells, and promoted viability under oxidative stress, although neither EPO nor ARA290 increased proliferation (O'Leary, Canning et al. 2019). EPO but not ARA290 enhanced endothelia cell tube formation in culture while in vivo administration in mice of ARA290, but not EPO, together with intravitreal injection of endothelial colony forming cells promoted their vasoreparative potential (O'Leary, Canning et al. 2019). ARA290 can attenuate inflammatory response and reduce the inflammatory retinal microenvironment. It is uncertain if ARA290 can mobilize and recruit other vasoactive cells into the ischemic retina.

Johnson, D. L., F. X. Farrell, F. P. Barbone, F. J. McMahon, J. Tullai, K. Hoey, O. Livnah, N. C. Wrighton, S. A. Middleton, D. A. Loughney, E. A. Stura, W. J. Dower, L. S. Mulcahy, I. A. Wilson and L. K. Jolliffe (1998). "Identification of a 13 amino acid peptide mimetic of erythropoietin and description of amino acids critical for the mimetic activity of EMP1." Biochemistry 37(11): 3699-3710.

O'Leary, O. E., P. Canning, E. Reid, P. M. Bertelli, S. McKeown, M. Brines, A. Cerami, X. Du, H. Xu, M. Chen, L. Dutton, D. P. Brazil, R. J. Medina and A. W. Stitt (2019). "The vasoreparative potential of endothelial colony-forming cells in the ischemic retina is enhanced by cibinetide, a non-hematopoietic erythropoietin mimetic." Exp Eye Res 182: 144-155.

Reviewer 4 Report

Comments and Suggestions for Authors

This is a well written review article on erythropoietin and metabolism. The article is very informative. I recommend this manuscript for publication with minor revision.

Minor comments:

1.       References 6 and 69 appear to reference the same article. Please double check.

2.       Can the author please elaborate on the interplay between EPO-regulated lipolytic and lipogenic gene expression in relation to lipid metabolism? Specifically, what are the potential implications of upregulated lipolytic genes on lipid metabolism, and how might downregulated lipogenic genes impact lipid metabolism to enhance metabolic health?

Author Response

Comments 1: References 6 and 69 appear to reference the same article. Please double check.

Response 1: We thank the reviewer for the noticing the duplication. Ref. 69 has been corrected to Ref. 6.

Comments 2: Can the author please elaborate on the interplay between EPO-regulated lipolytic and lipogenic gene expression in relation to lipid metabolism? Specifically, what are the potential implications of upregulated lipolytic genes on lipid metabolism, and how might downregulated lipogenic genes impact lipid metabolism to enhance metabolic health?

Response 2: 

We thank the reviewer for this question and now include following in the text at the end of section 11.

EPO administration in mice increased oxidative metabolism and fatty acid oxidation and lipolytic gene expression, and promoted mitochondrial content and uncoupled respiration in adipose tissue [5, 6]. Increase expression of lipolytic genes increases triglyceride catabolism, breaking down stored fat in adipose tissue and numerous nonadipose tissues into free fatty acids and promoting lipid mobilization and utilization in other tissues such as energy production for skeletal muscle, and their effect on gene transcription, cell cycle and cell growth  [101]. These activities associated with increased lipid metabolism likely contribute to EPO activity in non-erythroid tissue including enhanced wound healing in skeletal muscle [42, 43].

Metabolism coregualtors, peroxisome proliferator-activated receptor (PPAR)α and Sirt1, and RUNX1 combine to mediate, in part, EPO regulation of metabolism including lipid metabolism associated gene expression [5, 6]. Decrease in fat mass in mice treated with EPO is consistent with EPO stimulated down regulation of lipogenic genes that decreases expression of enzymes crucial for fatty acid production, reduces fatty acid synthesis, lipid production and fat storage [6, 7]. EPO regulation reduction of lipogenic gene could potentially help reduce fat synthesis and associated with high triglyceride level in liver and adipose tissues in metabolic disease or obesity [102]. Together, EPO as a novel regulator of energy homeostasis including lipolytic and lipogenic gene expression provides a potential strategy to enhance metabolic health and protect against obesity.

Reviewer 5 Report

Comments and Suggestions for Authors

This review summarizes advances in understanding the role of EPO in metabolic regulation, apart from the erythroid function of EPO. The senior author is the preeminent authority in this area and the article contains much valuable information. My main concern is that the information is poorly organized. This becomes apparent when sequentially reading the bold headings. Background, phenomenologic descriptions and mechanistic insights are intermixed, making the article a less useful introduction to the subject than it could be. There are also stylistic problems detailed under Specific Comments.

I would suggest clearly separating the ideas that form: a) the background information (both animal studies and human clinical observations from EPO studies and clinical use, including briefly erythroid effects) followed by b) a review of non-erythroid metabolic effects of EPO (the phenomenology) followed by c) a review of the molecular mechanisms that mediate the metabolic effects, perhaps contrasting them with the pathways that mediate the erythroid effects. Each of these can be subdivided similarly to the current version.

Specific Comments:

1) The title is not representative of the content of the article

2) There are several run-on sentences e.g. line 106. Make sure that the ideas in each sentence are related to each other.

3) CNS protection (lines 109-113) should be presented in the background section or connected better to metabolic mechanisms

4) The headings are not always appropriate. Line 113: Clinical trials  are mentioned under the heading: "3. Lessons learned from EPO-EPOR response in animal models"

5) The heading:  "5. EPO treatment of glucose metabolism" implies that EPO is used for treatment of glucose metabolism. Perhaps "Effects of EPO on glucose metabolism" or something similar would be a better choice.

6) Line 166: EPO effects on BAT are mentioned without explanation.

7) Line 213: Run-on sentence with a dangling modifier: "In contrast, increased EPO through transgenic expression, EPO 213 administration, or hypoxia promoted a lean phenotype and improved glucose tolerance, including increased EPO in the brain via transgenic expression or intracerebroventricular  pump implantation, which does not cross the blood-brain barrier to increase erythropoiesis [67, 72, 73]."

8) There are missing or extra words in several places, e.g. line 55, 241

9) Ideas are repeated e.g. line 188 and line 214

10) Please change "EPO treatment also contributed to improved cognitive dysfunction associated with hippocampal neurodegeneration in rodent models of diabetes [78, 79]."

11) Two sections, 8 and 10 deal with ARA290. These could be combined. 

12) References to Figure 2 in section 9 seem to refer to a nonexistent figure.

13) Figure 2 belongs to section 11.

Author Response

Comments 1: This review summarizes advances in understanding the role of EPO in metabolic regulation, apart from the erythroid function of EPO. The senior author is the preeminent authority in this area and the article contains much valuable information. My main concern is that the information is poorly organized. This becomes apparent when sequentially reading the bold headings. Background, phenomenologic descriptions and mechanistic insights are intermixed, making the article a less useful introduction to the subject than it could be. There are also stylistic problems detailed under Specific Comments.

I would suggest clearly separating the ideas that form: a) the background information (both animal studies and human clinical observations from EPO studies and clinical use, including briefly erythroid effects) followed by b) a review of non-erythroid metabolic effects of EPO (the phenomenology) followed by c) a review of the molecular mechanisms that mediate the metabolic effects, perhaps contrasting them with the pathways that mediate the erythroid effects. Each of these can be subdivided similarly to the current version.

Response 1: We thank the reviewer for helpful suggestions. The text has been modified accordingly.

Comments 2: 1) The title is not representative of the content of the article

Response 2: Title has been changed in consideration of the reviewer’s suggestion.

Commnets 3: 2) There are several run-on sentences e.g. line 106. Make sure that the ideas in each sentence are related to each other.

Response 3: The sentence has been corrected and the manuscript was reviewed with greater attention to language.

Comments 4: 3) CNS protection (lines 109-113) should be presented in the background section or connected better to metabolic mechanisms

Response 4: The text has been moved to a location earlier in the text describing EPO activity in brain.

Comments 5: 4) The headings are not always appropriate. Line 113: Clinical trials  are mentioned under the heading: "3. Lessons learned from EPO-EPOR response in animal models"

Response 5: Headings and text have been modified in consideration of the reviewer’s suggestion.

Comments 6: 5) The heading:  "5. EPO treatment of glucose metabolism" implies that EPO is used for treatment of glucose metabolism. Perhaps "Effects of EPO on glucose metabolism" or something similar would be a better choice.

Response 6: The heading has been changed as suggested.

Comments 7: 6) Line 166: EPO effects on BAT are mentioned without explanation.

Response 7: The text has been modified according to the reviewer’s suggestion: Other tissues that contribute to EPO response in improving obesity and glucose homeostasis in obese mice include brown adipose tissue (BAT) in which EPO increased interscapular fat mass, temperature, and secretion of FGF21, a metabolic regulator of glucose homeostasis and insulin sensitivity.

Comments 8: 7) Line 213: Run-on sentence with a dangling modifier: "In contrast, increased EPO through transgenic expression, EPO 213 administration, or hypoxia promoted a lean phenotype and improved glucose tolerance, including increased EPO in the brain via transgenic expression or intracerebroventricular  pump implantation, which does not cross the blood-brain barrier to increase erythropoiesis [67, 72, 73]."

Response 8: The text was modified accordingly: In contrast, increased EPO through transgenic expression, EPO administration, or hypoxia promoted a lean phenotype and improved glucose tolerance [69, 73, 74].

Comments 9: 8) There are missing or extra words in several places, e.g. line 55, 241

Response 9: The manuscript was reviewed with greater attention to language.

Comments 10: 9) Ideas are repeated e.g. line 188 and line 214

response 10: Repetition has been removed as suggested.

Comments 11: 10) Please change "EPO treatment also contributed to improved cognitive dysfunction associated with hippocampal neurodegeneration in rodent models of diabetes [78, 79]."

Response 11: This sentence has been moved to the section: EPO activity beyond erythropoiesis.

Comments 12: 11) Two sections, 8 and 10 deal with ARA290. These could be combined. 

Response 12: The two sections that deal with ARA290 have been combined as suggested.

Comments 13: 12) References to Figure 2 in section 9 seem to refer to a nonexistent figure.

Response 13: We appreciate the reviewer’s attention to detail and have deleted these references to Figure 2.

Comments 14: 13) Figure 2 belongs to section 11.

Response 14: Placement of Figure 2 has been corrected.

Round 2

Reviewer 5 Report

Comments and Suggestions for Authors

This is a very useful review of the nonerythroid effects of erythropoietin, with emphasis on lipid metabolism. After revision, the organization of the manuscript is greatly improved.